# Feasibility of the "Preventing functional decline in acutely hospitalized older patients (PREV_FUNC)" study—A three-armed randomized controlled pilot trial

Linda Sandberg[1,2]*, Anne-Marie Boström[3,4,5], Maria Hagströmer[1,6], Charlotte Lindgren[2], Miia Kivipelto[3,4,7,8], Christina Sandlund[1,6,9], Anna-Karin Welmer[1,10,11]

1 Division of Physiotherapy, Department of Neurobiology, Care Sciences and Society, Karolinska Institutet, Stockholm, Sweden, 2 Department of Geriatric Medicine, Capio Geriatrik Dalen, Capio Elderly and Mobil Care, Stockholm, Sweden, 3 Theme Inflammation and Aging, Nursing Unit Aging, Karolinska University Hospital, Huddinge, Sweden, 4 Research and Development Unit, Stockholms Sjukhem, Stockholm, Sweden, 5 Division of Nursing, Department of Neurobiology, Care Science and Society, Karolinska Institutet, Huddinge, Sweden, 6 Academic Primary Health Care Centre, Region Stockholm, Stockholm, Sweden, 7 Division of Clinical Geriatrics, Department of Neurobiology, Care Sciences and Society, Karolinska Institutet, Stockholm, Sweden, 8 Institute of Public Health and Clinical Nutrition, University of Eastern Finland, Kuopio, Finland, 9 Division of Family Medicine and Primary Care, Department of Neurobiology, Care Sciences and Society, Karolinska Institutet, Stockholm, Sweden, 10 Women´s Health and Allied Health Professionals Theme, Medical Unit Medical Psychology, Karolinska University Hospital, Stockholm, Sweden, 11 Aging Research Center, Department of Neurobiology, Care Sciences and Society, Karolinska Institutet and Stockholm University, Stockholm, Sweden

* linda.m.sandberg@ki.se

## Abstract

### Background

Recent studies indicate that in-hospital exercise can mitigate the risk of functional decline in acutely hospitalized older adults. However, there is a lack of studies that compare different types of exercise interventions. This feasibility study was conducted in preparation for a three-armed randomized controlled trial. The aim was to examine the process feasibility (in terms of recruitment and retention rate, intervention compliance and acceptability), and scientific feasibility (in terms of presence of adverse events, and trends with 95% confidence intervals of the outcome measures) of the trial.

### Methods

Patients aged $\geq$75 years, were included from geriatric medical wards at three hospitals in Stockholm, Sweden. Participants in two groups received a specialized intervention program, i.e., Simple or Comprehensive exercise program, respectively and one group received usual care. Assessments were conducted at hospital admission and discharge, and data were analyzed with descriptive statistics.

### Results

In the spring 2022, 63 patients met the inclusion criteria and 39 accepted to participate (recruitment rate: 61.9%). COVID-19 affected the inclusion period. A total of 33 participants

**Data Availability Statement:** The data in the study are pseudonymized (coded) personal data, and European General Data Protection Regulation

**Funding:** The PREV_FUNC trial is supported by grants provided by the Swedish Research Council for Health, Working Life and Welfare, Grant Number 2021-01788; Region Stockholm (ALF project), Grant Numbers FoUI-960460, and FoUI-972778, Center for Innovative Medicine (CIMED), Region Stockholm, Grant Number FoUI-961330; Karolinska University Hospital; Konung Gustaf V:s och Drottning Victorias Frimurarestiftelse (PI AKW) and Aleris Healthcare, Grant Number 2021-5 (LS). The funders have no role in design of the trial, collection, management, analysis, interpretation of data, writing manuscripts, or in the decision to submit for publication.

**Competing interests:** The authors have declared that no competing interests exist.

completed the study (i.e., were assessed at baseline and discharge, retention rate: 84.6%). Participants in the Simple and the Comprehensive exercise programs performed 88.9% and 80% of the possible training sessions, respectively. Both interventions were accepted by the participants and no adverse events were reported. The intervention groups showed a higher median change from admission to discharge than the control group on the Short Physical Performance Battery, the main outcome measure of the trial.

## Conclusion

The result of this pilot study suggests that the trial design is feasible and potentially useful for preventing functional decline in acutely hospitalized older adults. A full-scale trial will, however, require some considerations with respect to routines and logistics. The trial was registered at ClinicalTrials.gov, 4 May 2022, registration number NCT05366075.

## Introduction

Hospitalization because of acute medical illness is associated with several negative health consequences in older adults, such as physical functional decline, reduced well-being, and increased risk of readmission and death after discharge [1–3]. At discharge, more than one-third of hospitalized older persons have lost their ability to perform one or more basic activities of daily living independently [1]. These health consequences seem at least in part to be due to the hospitalization itself rather than the illness that caused the admission [4].

While intended to bring benefits, a hospital stay is often accompanied by almost total physical inactivity [5, 6]. Such excessive time spent in bed may lead to an accelerated development of sarcopenia, frailty, and physical function decline [5, 7, 8]. Physical exercise plays an essential role in preventing these health consequences in older hospitalized patients [9, 10]. A meta-analysis of Randomized Controlled Trials (RCTs) concluded that 25–50 minutes a day of slow-paced walking or multicomponent exercise at the hospital can improve functional capacity and reduce the number of adverse events for older adults [10]. It can however be challenging to find feasible interventions for older people with limited physical capacity [11]. Furthermore, there is a lack of studies that compare different types of exercise interventions. Such a comparison is important since different interventions might not provide the same effect [12].

Previous research has shown promising results in reversing functional decline for training programs in the context of acute-care units at hospitals in Spain. One is a simple multicomponent exercise program studied by Ortiz-Alonso et al. [11], while the other is a comprehensive multicomponent exercise program investigated by Martínez-Velilla et al. [13]. These programs were evaluated through RCTs involving individuals aged 75 years and older who were acutely hospitalized. The simple exercise program consisted of walking and sit-to-stand exercises [11]. The comprehensive exercise program included balance, moderate-intensity resistance, and walking exercises [13].

Based on the need to find the most effective exercise intervention for older adults and considering that a simple exercise program may be more feasible if proven effective due to time constraints in acute care settings, we designed the "Preventing functional decline in acutely hospitalized older patients (PREV_FUNC)" study. The aim of the PREV_FUNC study is to examine 1) if multicomponent exercise interventions (interventions that include both mobility and strengthening exercises) have effects on physical function compared to usual care in older

adults, and 2) if a comprehensive multicomponent exercise program is more effective than a simple multicomponent exercise program that only include walking and sit-to-stand exercises. The details of the PREV_FUNC study are described in a published protocol [14].

Since the study design of the PREV_FUNC study is untested, our intention with the current feasibility study was to increase the understanding of the project to improve the design and enhance the likelihood of success when conducting a larger trial [15, 16]. Furthermore, transparent reporting of a project's feasibility can also be useful to other researchers who plan similar studies [16]. The aim of this study was therefore to examine the process feasibility (in terms of recruitment and retention rate, intervention compliance and intervention acceptability), and scientific feasibility (in terms of presence of adverse events, trends with 95% confidence intervals [CIs] of the outcome measures, including physical function, activities of daily living [ADL], health-related quality of life [HRQL], and sarcopenia at hospital discharge) of the planned PREV_FUNC study.

## Materials and methods

### Study design and participants

The present study was a feasibility study in preparation for a full-scale RCT. The trial was designed as a three-armed, single blind, block randomized controlled multi-center trial. Participants were included consecutively during weekdays in one of the three groups in a time-dependent manner; they were recruited to one group at the time, in blocks of three weeks to reduce the risk of the groups influencing each other [11]. The order of the blocks was randomized by the researchers. Due to the COVID-19 pandemic the start of the study was delayed from autumn 2021 to spring 2022. Participants were recruited from March 15th to June 3rd in 2022, from three hospitals (two geriatric clinics and a geriatric department at a university hospital) in the Stockholm region. The patients could come directly from home, or via an emergency department. Due to the pilot nature of the study, no formal sample size calculation was required. The total number of included participants was lower than initially planned, which was 24–30 per hospital. However, based on comparisons with previous studies and recommendations by Whithead et al. [17], a total sample size of 30 participants is considered sufficient.

The inclusion criteria were age ≥75 years, ability to stand up from a sitting position independently or with minimal help from one person, and ability to communicate and collaborate with the research staff. The exclusion criteria were terminal illness, major medical condition that contraindicates exercise, living in nursing home, or previous inclusion in the pilot trial. A study protocol for the full-scale trial has been registered at ClinicalTrials.gov, 4 May 2022, registration no. NCT05366075, https://clinicaltrials.gov/study/NCT05366075. Reporting of the current study follows the Consolidated Standards of Reporting Trials statement: Extension to randomized pilot and feasibility trials [15]. Please see S1 Checklist.

### Ethics statement

Ethical approval was obtained from *The Swedish Ethical Review Authority* Dnr. 2020–06505, and 2021-06788-02. Please see S1–S6 Files. The research within this project has been conducted according to the principles expressed in the Declaration of Helsinki. All participants received verbal and written information and informed written consent was obtained before inclusion. The collected data were pseudonymized and stored in accordance with the General Data Protection Regulation. Only the responsible researchers had full access to the data.

## Procedure

Incoming patients were screened according to the study criteria and eligible patients were invited to participate within 36 hours of admission. Consent was obtained by research staff with sensitivity, meaning that no one was enrolled in the study if they appeared to have difficulty understanding what was meant by informed consent or expressed discomfort regarding participating. Physiotherapists and occupational therapists, specifically trained for the study and blinded to group allocation, assessed physical function, ADL, HRQL, and sarcopenia at baseline (hospital admission) and discharge. At baseline, data were also collected on previous falls. The baseline assessment was conducted as soon as possible after the patient had accepted to participate in the study and was carried out in a separate space to protect the privacy of the patient. The interventions started the same day as the baseline assessment, or the day after, and was performed on all weekdays until discharge. Due to shortage of staff, the interventions were not performed on weekends as initially planned. All participants in the study received usual care so the interventions were adjunctive.

## Interventions

The study included two interventions: Simple exercise program and Comprehensive exercise program. Both interventions were adapted to each participant's capacity.

The Simple exercise program was adapted from an intervention by Ortiz-Alonso, et al. [11] and included four sessions per day with a total duration of 20–30 minutes. It consisted of sit-to stand-exercises (10 repetitions in 3 sets) and walking for the remaining part of each session. The exercises were performed in the patient's room and the corridor of the ward. The intervention was modified from the original program [11], by increasing the number of sessions from three to four and thus the total duration of the exercise time from 20 minutes to 20–30 minutes per day. This was done to make the two interventions more comparable in terms of daily exercise time.

The Comprehensive exercise program was adapted from a multicomponent exercise intervention by Martinez-Velilla, et al. [13]. It consisted of two daily sessions: one in the morning and one in the afternoon. The total duration was 40 minutes (i.e., 20 minutes per session). The morning session included progressive resistance, balance, and walking exercises. The resistance training was focused on lower-extremity muscles and weight cuffs were used. Balance and gait exercises included for example line walking, stepping practice, and walking a path with small obstacles. The afternoon session consisted of functional exercises using light loads, such as knee extension and flexion, hand training with a ball, and daily walking exercises. The difference to the program by Martinez-Velilla, et al. [13] was that weight cuffs and resistance bands were used for the resistance exercises instead of resistance exercise machines.

The Control group received usual care, which is based on teamwork including physiotherapy. The physiotherapist in the team focuses on mobility assessments.

All participants had an individual logbook. In the logbook, and reasons why a session was not completed were registered by the research staff, for participants in the intervention groups. Adverse events such as falls during the hospital stay were registered for all study participants.

## Primary outcome measures (process feasibility)

The primary outcome was process feasibility, which included recruitment and retention rate, intervention compliance and intervention acceptability [16]. Recruitment rate was defined as the ratio between the number of eligible and included participants. Retention rate was defined as the proportion of patients that remain to follow-up between the two assessments (baseline and discharge). Intervention compliance was defined as the proportion of completed exercise

sessions of the possible sessions. Intervention acceptability was assessed at the follow-up session by a survey. The survey contained nine questions with three or five ordered response options and an open-ended question of how the participants experienced the exercise. For the analysis, the five response options were combined into three groups: *Partially agree* and *Agree to a great extent* were merged to *Agree*; *Disagree* and *Totally disagree* were merged to *Disagree;* and *Neither agree nor disagree* was defined as *Neutral*. To assess the feasibility, the findings were compared to other exercise interventions on acutely hospitalized older adults and needs for potential modifications for the full-scale trial were discussed in the research group.

## Secondary outcome measures (scientific feasibility)

The secondary outcome was scientific feasibility, focusing on presence of adverse events, and trends with 95% CIs of the outcome measures. Adverse events were assessed through registration in the participants' logbooks. Data collected for the outcome evaluation included physical function, ADL, HRQL, and sarcopenia, assessed at baseline (hospital admission) and discharge. Physical function was assessed with the Short Physical Performance Battery (SPPB) [18], which will be the primary outcome of the full-scale trial. The SPPB consists of three components: standing balance, walking speed, and chair stand test. It is sensitive to change and has been shown to predict a wide range of clinical outcomes in frail older adults [19, 20]. ADL was assessed with Barthel Index [21] and health-related quality of life with the EuroQol–5 Dimension (EQ-5D-3L) [22, 23]. Due to few participants in the last category (reporting severe problems), the second (moderate problems) and last categories were combined in the analysis. Sarcopenia was defined according to the European Working Group's revised criteria on Sarcopenia in Older People [24] and included muscle strength and muscle mass. Muscle strength was evaluated by testing handgrip strength twice in each hand and the best overall result was used in the analysis. Low muscle strength was considered as handgrip of <27 kg for men and <16 kg for women [24]. Low muscle mass was defined as having a calf circumference of <34 cm for men and <33 cm for women [25]. We defined "probable sarcopenia" as having low muscle strength and normal muscle mass and "sarcopenia" as having low muscle strength and low muscle mass [24]. Due to time constraints, data on cognition and nutritional supplementation were not collected as initially planned.

## Demographic and clinical data

Data on demographic and clinical characteristics were collected from patient records and included: age, sex, cohabitation status, reason for hospital admission, comorbidities, frailty status according to the Clinical Frailty Scale [26], nutritional risk according to the Mini Nutritional Assessment-Short Form, MNA-SF [27], length of hospital stay and discharge destination.

## Data analysis

Recruitment rate and retention rate, intervention compliance, acceptance, demographic and clinical data and baseline values of the outcome measures were presented as numbers and proportions (%), or medians and interquartile ranges. The written responses to the open-ended question were summarized. To estimate the trends of the treatment effect, we first calculated change from baseline to discharge in each outcome measure by subtracting the baseline score from the discharge score. The SPPB and Barthel Index were employed as continuous scales in univariate quantile regression models to assess median change by group in each outcome measure with 95% CI. The CI was utilized to describe the uncertainty associated with the point estimates for trend. For categorical variables (sarcopenia and EQ-5D-3L), each variable was

categorized as worsening, stable or improvement, and then the proportions in each category was calculated with 95% CI. Statistical analyses were performed using Stata version 17 (Stata Corp., College Station, TX).

## Results

### Process feasibility

**Recruitment and retention rate.** During the study period, 63 patients were eligible to participate and 42 accepted to participate. Of these, three participants were lost before the baseline assessment, thus 39 people were included in the study, corresponding to a 61.9% recruitment rate. Six participants dropped out from the study, two from each group. A total of 33 participants completed the study (i.e., were assessed at baseline and discharge), corresponding to a retention rate of 84.6% between the two assessments (Fig 1). The reasons for drop out were medical deterioration, staff shortages, and rapid and unscheduled discharge. The proportion of retention differed between the three hospitals; the university hospital had a higher proportion of drop out (41.7%), while the two geriatric clinics had a lower proportion (5.3% and 11.1%). When communicating with the staff, it appeared that the COVID-19 pandemic affected both the recruitment and retention rate. For instance, the research staff had to wait for a negative covid test before a patient was invited to participate. Depending on how quickly the response samples were delivered the timeline to include patients before 36 hours sometimes expired.

**Clinical and demographic characteristics of the participants who completed the study.** Of the 33 participants who completed the study, the median age was 86.0 years (IQR 80.0–92.0), 20 (60.6%) were women, and 22 (66.7%) were living alone. The main cause for admission to the hospital were respiratory and circulatory disorders (n = 11, 33.3%), musculo-skeletal disorders and injuries (n = 7, 21.2%), general symptoms/deconditioning (n = 7, 21.2%), and genitourinary and infectious disorders (n = 6, 18.2%). For the 6 participants who dropped out of the study, the median age was 81.3 years (IQR 77.0–87.0), 2 were women and 4 were living alone.

Nine participants (27.3%) reported no falls in the previous 12 months, 9 (27.3%) had fallen once, and 15 (45.5%) had fallen on two or more occasions. At baseline, the median score for the Clinical Frailty Scale in the total sample was 5 (IQR 3.5–6.0) (possible range 1–9, with a higher score indicating worse frailty). The proportion of participants at risk of malnutrition at baseline was 66.7% (i.e., scoring ≤11 of 14 on the MNA-SF) and the median length of hospital stay was seven days. Most participants were discharged to their home, but one participant was discharged to a nursing home and one to another geriatric clinic. For a description of the clinical and demographic characteristics of the study population by subgroups, see Table 1.

**Intervention compliance.** The median number of completed training sessions for the participants in the Simple exercise program was 10.0 (IQR 6.0–11.0), corresponding to 88.9% (IQR 55.6–100%) of the possible training sessions. For the Comprehensive exercise program, the median number of completed training sessions was 4 (IQR 3.0–6.0), corresponding to 80% (IQR 60.0–88.9%) of the possible training sessions. Reported reasons for not completing a training session included pain, isolation due to infection, booked contact with another healthcare provider, affected by sleep medication, and shortage of staff.

**Intervention acceptability.** Most participants (93.8%) agreed that they received enough information about the study (Table 2). In the intervention groups, all but one agreed that they received enough information during the training sessions and all agreed that it felt meaningful to exercise at the hospital and that they were motivated to exercise. All but one in the Comprehensive exercise group rated the exercise as relevant. The percentage who agreed that they

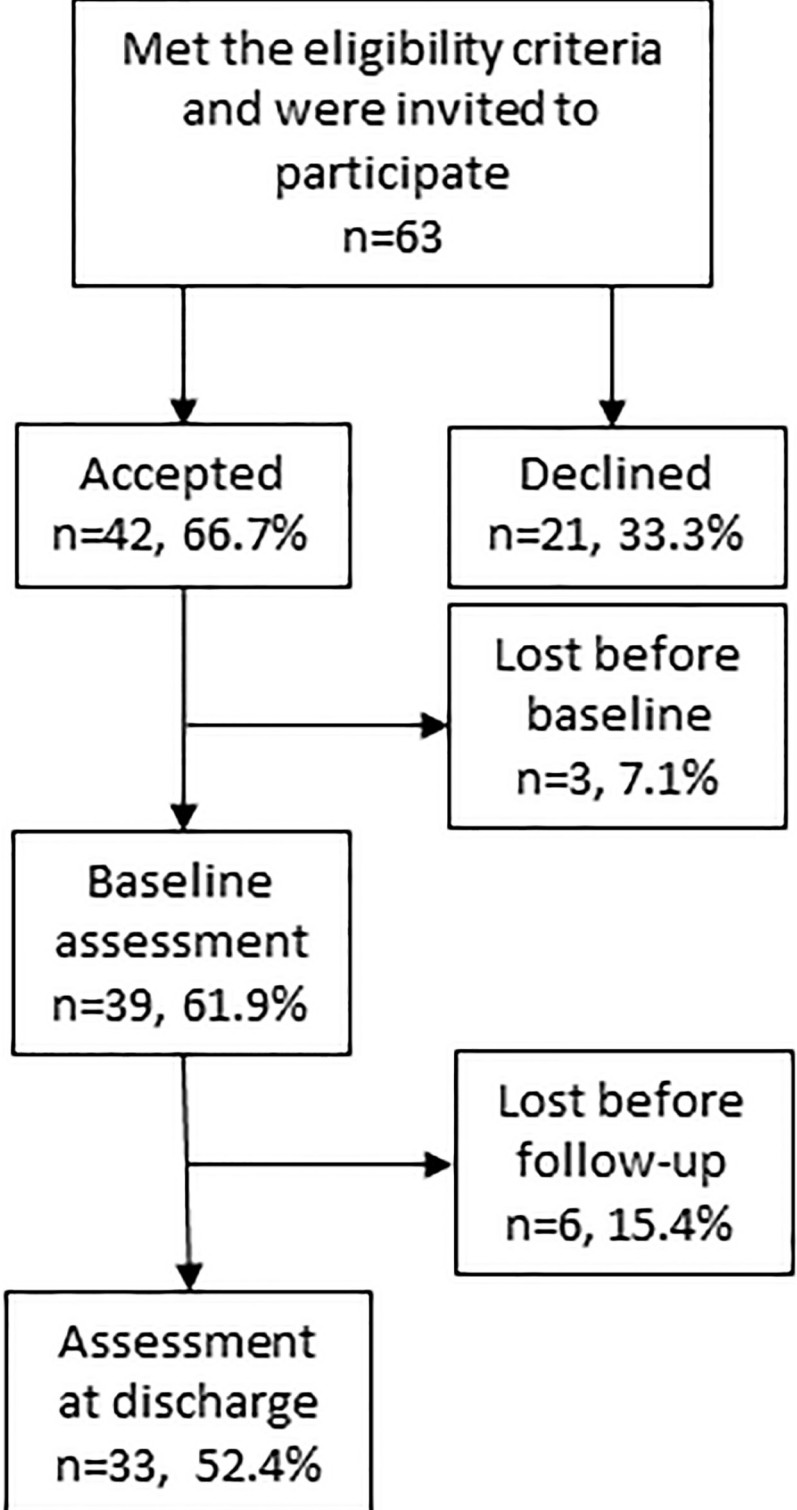

**Fig 1. Flowchart of the PREV_FUNC pilot trial.**

**Table 1. Characteristics of the study population by group, n = 33.**

| Characteristic | Simple exercise program | Comprehensive exercise program | Control group |
|---|---|---|---|
| | n = 11 | n = 14 | n = 8 |
| **Demographic data** | | | |
| Age, Median (IQR) | 84 (80–93) | 83 (77–90) | 90 (85–94) |
| Women, n (%) | 6 (54.5) | 9 (64.3) | 5 (62.5) |
| Living alone, n (%) | 2 (18.2) | 5 (35.7) | 4 (50.0) |
| **Clinical data** | | | |
| Main cause of admission, n (%) | | | |
| Respiratory and circulatory disorders | 2 (18.2) | 4 (28.6) | 5 (62.5) |
| Musculoskeletal disorders and injuries | 4 (36.4) | 2 (14.3) | 1 (12.5) |
| Genitourinary and infectious disorders | 2 (18.2) | 2 (14.3) | 2 (25.0) |
| General symptoms/deconditioning | 2 (18.2) | 5 (35.7) | 0 (0) |
| Previous falls, n (%) | | | |
| 0 | 4 (36.4) | 3 (21.4) | 2 (25.0) |
| 1 | 1 (9.1) | 5 (35.7) | 3 (37.5) |
| ≥2 | 6 (54.5) | 6 (42.9) | 3 (37.5) |
| CFS at baseline, Median (IQR) | 5 (3–6) | 5 (3–6) | 5 (5–6) |
| Risk of malnutrition at baseline, n (%) | 7 (70.0) | 6 (60.0) | 5 (71.4) |
| Length of hospitalization, Median (IQR), days | 6 (5–8) | 7 (4–10) | 7.5 (5–9.5) |
| **Main outcome measure** | | | |
| SPPB at baseline, Median (IQR) | 3.5 (2–7) | 4.5 (2–6) | 5 (3–5.5) |
| **Secondary outcome measures** | | | |
| Barthel Index at baseline, Median (IQR) | 80 (50–95) | 82.5 (70–95) | 70 (57.5–85) |
| Sarcopenia at baseline, n (%) | | | |
| No sarcopenia | 7 (63.6) | 8 (57.1) | 1 (14.3) |
| Probable sarcopenia | 2 (18.2) | 2 (14.3) | 4 (57.1) |
| Sarcopenia | 2 (18.2) | 4 (28.6) | 2 (28.6) |
| EQ-5D-3L at baseline, Any problems, n (%) | | | |
| Mobility | 8 (72.7) | 7 (50.0) | 4 (50.0) |
| Self-care | 3 (27.3) | 3 (21.4) | 3 (37.5) |
| Usual activities | 8 (72.7) | 8 (57.1) | 3 (37.5) |
| Pain/discomfort | 9 (81.8) | 9 (64.3) | 4 (50.0) |
| Anxiety/depression | 3 (27.3) | 6 (42.9) | 3 (37.5) |

CFS = Clinical Frailty Scale, SPPB = Short Physical Performance Battery, EQ-5D-3L = EuroQol–5 Dimension, IQR = Interquartile Range

Missing: Main cause of admission = 2, CFS = 4, Risk of malnutrition = 6, Length of hospitalization = 2, SPPB = 1, Sarcopenia = 1

could challenge themselves during exercise was higher in the intervention groups (Simple exercise program 90.9%; Comprehensive exercise program 85.7%) compared to the Control group (33.3%). One participant in the group receiving the Simple exercise program and two in the group receiving the Comprehensive exercise program rated the exercise intensity as too high.

In response to the question asking for their experiences of exercise during the hospital stay, participants in the Simple exercise group expressed that it was important to get support from a person: "It works better when you have someone helping you. You need a little help to get started". Participants stated that the training broke the monotony and led to social interactions. Same participants experienced the Simple exercise as demanding, although they expressed that it was worth the effort. However, some participants had wished for a more

**Table 2. Participants' acceptance of participating in the study, n = 33.**

| | Simple exercise program | Comprehensive exercise program | Control group |
|---|---|---|---|
| | **n = 11** | **n = 14** | **n = 8** |
| Did you get enough information about the study? n (%) | | | |
| Yes | 10 (90.9) | 13 (92.9) | 7 (100) |
| No | 1 (9.1) | 1 (7.1) | 0 (0) |
| I got enough information about the exercise from my physiotherapist/instructor, n (%) | | | |
| Agree | 11 (100) | 12 (93.3) | 5 (71.4) |
| Neutral | 0 (0) | 1 (7.7) | 1 (14.3) |
| Disagree | 0 (0) | 0 (0) | 1 (14.3) |
| It felt meaningful to exercise at the hospital, n (%) | | | |
| Agree | 11 (100) | 14 (100) | 4 (57.1) |
| Neutral | 0 (0) | 0 (0) | 2 (28.6) |
| Disagree | 0 (0) | 0 (0) | 1 (14.3) |
| I was motivated to carry out the exercise, n (%) | | | |
| Agree | 11 (100) | 14 (100) | 6 (85.7) |
| Neutral | 0 (0) | 0 (0) | 1 (14.3) |
| Disagree | 0 (0) | 0 (0) | 0 (0) |
| The content of the exercise was relevant, n (%) | | | |
| Agree | 11 (100) | 13 (92.9) | 6 (100) |
| Neutral | 0 (0) | 0 (0) | 0 (0) |
| Disagree | 0 (0) | 1 (7.1) | 0 (0) |
| I felt safe during the exercise, n (%) | | | |
| Agree | 11 (100) | 14 (100) | 6 (100) |
| Neutral | 0 (0) | 0 (0) | 0 (0) |
| Disagree | 0 (0) | 0 (0) | 0 (0) |
| I was able to challenge myself during the exercise, n (%) | | | |
| Agree | 10 (90.9) | 12 (85.7) | 2 (33.3) |
| Neutral | 0 (0) | 0 (0) | 1 (16.7) |
| Disagree | 1 (9.1) | 2 (14.3) | 3 (50.0) |
| The exercise intensity was. . ., n (%) | | | |
| Too low | 1 (9.1) | 2 (14.3) | 1 (20.0) |
| Moderate | 9 (81.8) | 10 (71.4) | 4 (80.0) |
| Too high | 1 (9.1) | 2 (14.3) | 0 (0) |
| It felt good to fill in the questionnaires, n (%) | | | |
| Agree | 11 (100) | 13 (100) | 5 (83.3) |
| Neutral | 0 (0) | 0 (0) | 1 (16.7) |
| Disagree | 0 (0) | 0 (0) | 0 (0) |

varied exercise program. In the group that was allocated to the Comprehensive exercise, one participant responded: "I thought it was liberating, and that it felt nice to exercise". Several participants emphasized the importance of getting out of bed to maintain physical fitness. Moreover, participants expressed that the exercise was adapted to their ability, which they found positive. For one participant, the exercise did not correspond to her expectations, since she thought that the training should have taken place at a gym. Some of the participants in the Control group expressed that they had strived to perform some exercise by themselves, e.g., to walk in the ward corridor. Another participant got support to practice stair training, which felt important before going home.

## Scientific feasibility

**Presence of adverse events.** No adverse events were reported during any of the exercise sessions. All participants agreed that they felt safe while exercising (Table 2).

**Trends of the outcome measures.** At baseline, participants in the intervention groups had somewhat lower scores on the main outcome, SPPB (median score 3.5 for the Simple exercise program and 4.5 for the Comprehensive exercise program), than those in the Control group (median score 5) (possible range 0–12, with a higher score indicating better physical function). Participants in the intervention groups had however somewhat higher baseline scores on the Barthel Index (higher score indicates better ADL ability) and higher prevalence of probable sarcopenia than those in the Control group (Table 1).

Results concerning changes in the outcome measures from baseline to discharge are presented in Table 3. A trend towards a greater improvement of the main outcome for participants in the intervention groups appeared; the median change score for SPPB for the Simple

**Table 3. Change from baseline (admission) to discharge in primary and secondary outcome measures by group, n = 33.**

| | Simple exercise program | Comprehensive exercise program | Control group |
|---|---|---|---|
| | n = 11 | n = 14 | n = 8 |
| **Main outcome measure** | | | |
| SPPB, Median change (95% CI) | 1 (0.0–2.0) | 2 (1.2–2.8) | 0 (-1.2–1.2) |
| **Secondary outcome measures** | | | |
| Barthel Index, Median change (95% CI) | 5 (-0.9–10.9) | 10 (4.7–15.3) | 5 (-2.0–12.0) |
| Sarcopenia, change from baseline (proportion of participants, 95% CI) | | | |
| Worsening | 18.2 (4.3–52.4) | 7.7 (1.0–41.2) | 16.7 (0.2–65.3) |
| Stable | 81.8 (47.6–95.7) | 61.5 (33.3–83.7) | 83.3 (34.7–97.9) |
| Improvement | 0 | 30.8 (11.5–60.3) | 0 |
| EQ-5D-3L, any problems, change from baseline (proportion of participants, 95% CI) | | | |
| Mobility | | | |
| Worsening | 0 | 28.6 (10.7–57.2) | 0 |
| Stable | 81.2 (47.7–95.7) | 64.3 (36.6–84.9) | 100 |
| Improvement | 18.2 (4.3–52.3) | 7.1 (0.9–39.0) | 0 |
| Self-care | | | |
| Worsening | 9.1 (1.2–45.9) | 14.3 (3.4–44.2) | 0 |
| Stable | 81.8 (47.7–95.7) | 71.4 (42.8–89.3) | 100 |
| Improvement | 9.9 (1.2–45.9) | 14.3 (3.4–44.2) | 0 |
| Usual activities | | | |
| Worsening | 9.1 (1.2–45.9) | 7.1 (0.9–39.0) | 28.6 (6.8–68.8) |
| Stable | 63.6 (32.8–86.3) | 85.7 (55.8–96.6) | 71.4 (31.2–93.2) |
| Improvement | 27.3 (8.6–59.9) | 7.1 (0.9–39.0) | 0 |
| Pain/discomfort | | | |
| Worsening | 0 | 0 | 28.6 (6.8–68.8) |
| Stable | 81.8 (47.7–95.7) | 78.6 (49.3–93.2) | 71.4 (31.2–93.2) |
| Improvement | 18.2 (4.3–52.3) | 21.4 (6.7–50.7) | 0 |
| Anxiety/depression | | | |
| Worsening | 9.1 (1.2–45.9) | 7.1 (0.9–39.0) | 14.3 (1.8–60.1) |
| Stable | 81.8 (47.7–95.7) | 85.7 (55.8–96.6) | 71.4 (31.2–93.2) |
| Improvement | 9.1 (1.2–45.9) | 7.1 (0.9–39.0) | 14.3 (1.8–60.1) |

SPPB = Short Physical Performance Battery, EQ-5D-3L = EuroQol–5 Dimension, CI = Confidence Interval

Missing: SPPB, control group = 1, EQ-5D-3L, control group = 1, Sarcopenia, control group = 2, comprehensive exercise program = 1

exercise program was 1 (95% CI 0.0–2.0) and 2 (95% CI 1.2–2.8) for the Comprehensive program, compared to 0 (95% CI -1.2–1.2) for the Control group (Table 3). The median change for the Barthel Index indicated a slightly greater improvement in favor of the Comprehensive program (10; 95% CI 4.7–15.3) compared to the Simple exercise program (5; 95% CI -0.9–10.9) and the Control group (5; 95% CI -2.0–12.0). Regarding sarcopenia, 30.8% (CI 11.5–60.3) of the participants in the Comprehensive exercise program improved, while no one in the other two groups showed improved results for sarcopenia.

Regarding, health-related quality of life (EQ-5D-3L), participants in the intervention groups reported both worsening and improvements between baseline and discharge, while those in the Control group remained more stable (Table 3). Participants in the Simple exercise program either reported being stable or having improvement of mobility and pain, and both worsening and improvements in all other items. Participants in the Comprehensive exercise program either reported being stable or having improvement of pain, and both worsening and improvements in all other items. The participants in the Control group reported being stable in mobility and self-care and being stable or having worsening of usual activities and pain. Only for anxiety/depression, participants in the Control group reported both worsening and improvements (Table 3).

## Discussion

This pilot study aimed to examine the process feasibility and scientific feasibility of the planned PREV_FUNC study. The results suggest that the trial design is feasible in terms of recruitment rate and retention rate, intervention compliance, intervention acceptability, and adverse events. Moreover, there was a positive trend for the main outcome measure of the PREV_FUNC study, SPPB, in favor of the intervention groups, suggesting a potential usefulness of the trial. Therefore, a full-scale RCT is considered suitable; however, there is potential for improvements as discussed below.

A review of recruitment and retention of participants in RCTs found that a median of 70% (IQR 51–87%) of eligible patients were included and a median 89% (IQR 79–97%) of randomized patients had valid primary outcome data for analysis [28]. The recruitment rate of 61.9% and retention rate of 84.6% in this study can be considered reasonable since acutely hospitalized older adults often live with frailty and have several medical conditions. Thus, it seems reasonable that around one third did not want to participate in an exercise study. Due to limited resource, we were however not able to register the total number of eligible participants, which limits our understanding of the recruitment rate. Even though the overall retention rate was considered reasonable the retention rate differed widely between the three hospitals. The university hospital had a higher drop out (41.7%), compared to the two geriatric clinics (5.3% and 11.1%), mainly due to logistical issues. Therefore, it must be considered whether it is possible to change something in our routines at the university hospital, e.g., in relation to staff resources, or if geriatric clinics are more suitable sites for conducting the full-scale trial. Since the result indicated that the COVID-19 pandemic affected both the recruitment and retention rate, this is another factor that needs to be considered when planning the time to include the desired number of participants in the full-scale trial which started in September 2022 and will be completed in 2024.

The study's intervention compliance can be considered acceptable. The intervention compliance for the Simple exercise program (89%) was higher compared to the study by Ortiz-Alonso et al., in which the participants completed on average 2/3 training sessions per day [11]. However, the intervention compliance for the Comprehensive exercise program (80%) was slightly lower compared to Martínez-Velilla et al. who had a compliance of 96% for the

morning sessions and of 83% for the evening sessions [13]. The Simple exercise program was led by several professions (physiotherapists, occupational therapists, and nursing assistants), while the Comprehensive exercise program was led only by physiotherapists. In communication with the staff, it was found that having several people involved in the exercise for each patient created logistical problems, e.g., unclear division of responsibilities. Therefore, in the full-scale trial the number of involved staff should be limited and there should be a person in charge of the trial at each hospital, which is also suggested by Kulmala et al. for successful implementation of an intervention [29]. It also appeared that it was important that the staff were motivated and had the resources to carry out their tasks of the study. Therefore, in the full-scale trial the project needs to generate commitment and have sufficient resources such as allocated time, which is also in line with Kulmala et al. [29].

In this study, the Comprehensive exercise program was modified from Martínez-Velilla et al. [13], with participants using weight cuffs and resistance bands instead of resistance exercise machines, and the Simple exercise program was adapted from Ortiz-Alonso et al. [11]. The staff who led the exercise programs communicated that these modified exercise programs overall worked well. They highlighted the strength of being able to adapt the exercises to each patient.

Overall, the participants found the exercise programs acceptable. All participants in the intervention group rated the exercise as meaningful, which is an important finding as a positive attitude towards exercise can contribute to better adherence [30]. However, the participants in this study were recruited to a study on exercise, therefore it is likely that they were more positive to exercise than the general population at geriatric clinics. Only half of the participants in the Control group rated the exercise as meaningful. It should however be considered that participants in the Control group may not have received any exercise during their hospital stay. Most participants rated the exercise intensity as moderate, while three participants in the intervention groups rated the exercise intensity as too high. This suggests that the exercise intensity is sufficient and it also emphasizes the importance of being responsive to the participants' experiences. It appeared that some participants in the Simple exercise program wished for a more varied exercise program. If proven effective, however, this program may be more feasible to implement in the time constrained acute care settings and may also be more suitable for self-training than the Comprehensive exercise program. Considering the shortened length of hospital stay for geriatric inpatients in Sweden [31], in the full-scale trial the participants in the Simple exercise program will be encouraged to continue with the exercises at home.

Our study did not incorporate a measure of cognitive function. Consequently, we cannot definitively conclude whether the intervention is feasible for older adults with cognitive impairment. Furthermore, we excluded individuals living in nursing homes for two specific reasons. Firstly, a comparable exercise program to the simple exercise program had already been tested in nursing homes as part of a previous study [32]. Secondly, considering that participants in the Simple exercise program will be encouraged to continue their exercises at home during the full-scale trial, it was deemed more appropriate to include only those living in their own homes.

No adverse events during the interventions were reported, which indicate that the interventions are safe. This finding is in line with previous research suggesting that staying active is safer than bed rest for acutely hospitalized older adults [10]. In this study we report preliminary trends with 95% confidence intervals of the outcome measures, as the aim of pilot studies is to examine feasibility, not effectiveness [16]. All the examiners received continuous training and support in how to perform the assessments correctly. The examiners instructed the participants not to tell them details about their training to remain blinded during the assessment.

Furthermore, a strength to our study is that we used standardized instruments to assess scientific feasibility, which reduces the risk of misclassification bias. Given the small sample size and the aim of feasibility studies, which is to assess the feasibility of a study rather than the effect of interventions, we made the decision not to present p-values for group differences. Instead, we utilized 95% CIs to describe the uncertainty associated with the point estimates for trend.

In line with the studies by Ortiz-Alonso and Martinez-Velilla, most of our outcomes showed trends in favor of the intervention groups [11, 13]. For example, the median change for the participants in the Simple and Comprehensive exercise program were one and two points on the SPPB, the main outcome measure of the PREV_FUNC study. This indicates a clinically meaningful difference of physical function in favor of the intervention groups [33]. In contrast to this finding, almost 30% of the group receiving the Comprehensive exercise program rated themselves as worse in Mobility on the EQ-5D-3L at discharge, while no one in the Simple exercise program and the Control group rated deterioration in Mobility. A possible reason for the discrepant results between the SPPB and EQ-5D-3L could be that the Comprehensive exercise program challenged the participants' ability more, which affected their experience of their own mobility. However, due to the limited sample size and the wide CIs, which indicate substantial uncertainty around the point estimates, the results need to be interpreted with caution. A promising finding is that no one in the intervention groups rated a worsening of Pain/discomfort on the EQ-5D-3L, and around 20% rated an improvement. By contrast, almost 30% of the participants in the Control group reported a worsening of Pain/discomfort and no one rated an improvement. This is in line with previous research suggesting that exercise and rehabilitation programs may reduce pain [34, 35].

## Conclusion

This pilot study suggests that the trial design of the PREV_FUNC study is feasible and that the interventions are potentially useful for preventing functional decline in acutely hospitalized older adults. Based on the experiences of participants and staff, the modified training programs can be continued to be carried out as in the pilot trial. Based on the results of this study, a full-scale RCT is suggested. A full-scale trial will, however, require the following considerations and improvements; 1) Given the high drop out at the university hospital, our routines at the university hospital need to be improved, or it should be considered if geriatric clinics are more suitable sites for conducting the full-scale RCT; 2) Considering that COVID-19 may still affect the recruitment and retention rates, it may take longer time to include the desired number of participants compared to similar trials before the pandemic; 3) The number of involved staff should be limited and there should be a person in charge of the trial at each hospital.

## Supporting information

**S1 Checklist. CONSORT extension pilot and feasibility.**
(DOC)

**S1 File. Ethical approval 1—In Swedish.**
(PDF)

**S2 File. Ethical approval 1—English translation.**
(DOCX)

**S3 File. Ethical approval 2—In Swedish.**
(PDF)

**S4 File. Ethical approval 2—English translation.**
(DOCX)

**S5 File. Ethical application—In Swedish.**
(PDF)

**S6 File. Ethical application—English translation.**
(PDF)

## Acknowledgments

A special thanks to all the participants who took part in this study. The authors also thank all the staff and managers who contributed and made it possible to include participants from the wards. We would also like to thank Kristina Dalin Eriksson for reviewing the language.

## Author Contributions

**Conceptualization:** Linda Sandberg, Anne-Marie Boström, Anna-Karin Welmer.

**Data curation:** Linda Sandberg, Anna-Karin Welmer.

**Formal analysis:** Linda Sandberg, Anna-Karin Welmer.

**Funding acquisition:** Linda Sandberg.

**Investigation:** Linda Sandberg, Anne-Marie Boström, Anna-Karin Welmer.

**Methodology:** Linda Sandberg, Anne-Marie Boström, Anna-Karin Welmer.

**Project administration:** Linda Sandberg, Anna-Karin Welmer.

**Resources:** Anna-Karin Welmer.

**Supervision:** Anna-Karin Welmer.

**Validation:** Linda Sandberg, Anna-Karin Welmer.

**Writing – original draft:** Linda Sandberg, Anna-Karin Welmer.

**Writing – review & editing:** Linda Sandberg, Anne-Marie Boström, Maria Hagströmer, Charlotte Lindgren, Miia Kivipelto, Christina Sandlund, Anna-Karin Welmer.

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
