## [Decision Letter · Decision Letter 0]

4 Mar 2024

PONE-D-23-42196Feasibility of the “Preventing functional decline in acutely hospitalized older patients (PREV_FUNC)” study – A three-armed randomized controlled pilot trialPLOS ONE

Dear Dr. Sandberg,

Thank you for submitting your manuscript to PLOS ONE. After careful consideration, we feel that it has merit but does not fully meet PLOS ONE’s publication criteria as it currently stands. Therefore, we invite you to submit a revised version of the manuscript that addresses the points raised during the review process.

Kind regards,

Sascha Köpke

Academic Editor

PLOS ONE

3. In the online submission form, you indicated that [Data Availability Statement: The data in the study are pseudonymised (coded) personal data, and Swedish legislation prohibits us from sharing this completely open. The dataset only includes data from 39 human research participants. Due to the small sample size risk of identification of individual participants exists if the data is shared, even though data is de-identified. Data is available upon request, and requests for access to the data can be put to our Research Data Office (rdo@ki.se) at Karolinska Institutet.]. 

Reviewers' comments:

Reviewer's Responses to Questions

**Comments to the Author**

1. Is the manuscript technically sound, and do the data support the conclusions?

Reviewer #1: Yes

Reviewer #2: Yes

2. Has the statistical analysis been performed appropriately and rigorously? 

Reviewer #1: Yes

Reviewer #2: Yes

3. Have the authors made all data underlying the findings in their manuscript fully available?

Reviewer #1: Yes

Reviewer #2: No

4. Is the manuscript presented in an intelligible fashion and written in standard English?

Reviewer #1: Yes

Reviewer #2: Yes

5. Review Comments to the Author

Reviewer #1: Manuscript PONE-D-23-42196

Feasibility of the “Preventing functional decline in acutely hospitalized older patients” study – A three-armed randomized controlled trial

This manuscript described a randomized controlled trial assessing recruitment and retention rates, intervention compliance, acceptability, and scientific feasibility of 3 different exercise programs for acutely ill older inpatients in geriatric units or clinics of 3 different hospitals in Sweden. The authors highlight that few studies have compared different exercise programs in the acute care setting among older patients.

I think overall this is a well-designed assessment and the authors did a very good job of writing up their findings. One question I have relates to inpatients with mild cognitive impairment (MCI). Other studies have shown that at least 1/3 of older inpatients have some level of cognitive impairment, and that patients with MCI are very limited in terms of physical function and mobility. Have the authors considered the impact of their interventions for patients with MCI? It seems these interventions are helpful for many but not all patients. Also, what happens when a patient develops delirium during hospitalization? How will the authors address this issue in the later RCT? The authors should explain why they are not assessing cognition at baseline using a standardized validated tool such as the Mini-cog. This is important not only for confirming the patient can provide written consent, but also for confirming cognition status. I think this piece of data should be prioritized during recruitment and enrollment for the later RCT and can be used to better understand the results of the study. If it is possible to include some patients with MCI, that would ultimately provide more evidence to present related to this trial. Alternatively, if the exercise programs are not appropriate for patients with MCI that should be clearly articulated and explained.

The first reference was cited twice – reference 1 and reference 3 in the bibliography.

Reviewer #2: This paper reports preliminary results from a feasibility study of in-hospital exercise. The manuscript is clearly written with intervention and analytical details well-documented. The major limitation is the small sample size. However, the authors clarified that the purpose of the study is just to examine the feasibility rather than intervention effectiveness. I only have a few minor suggestions:

1. The study excludes individuals living in nursing homes. It would be helpful if the authors could provide an explanation for this exclusion, given that the intervention is conducted in hospitals.

2. In several instances, the purpose of the study is stated as examining the trends and variance of the outcome measures Does “variance” here actually mean “change” as no variance was reported? If so, please consider revise as the readers may confuse it with the statistical term variance.

3. Considering the small sample size, univariate quantile regression may not have sufficient power. If the intention is to compare the mean or median of the SPPB, I recommend considering non-parametric tests such as the Kruskal-Wallis test.

4. When analyzing the SPPB and Barthel Index, will the sum scores be used? If so, please note that these scores are continuous rather than ordinal. Additionally, ordinal scales are technically a type of categorical variable. I suggest the authors review and clarify the definitions when describing the variables used for analysis.

5. The paper mentions that three participants were lost before the baseline assessment. It would be helpful to provide an explanation for the reasons behind this loss.

6. Although the sample size for each group is small, it is still worth discussing whether the differences in sample characteristics and outcome scores at baseline and after the intervention are statistically significant based on non-parametric tests.

6. PLOS authors have the option to publish the peer review history of their article (what does this mean?). If published, this will include your full peer review and any attached files.

Reviewer #1: **Yes: **Christine Loyd

Reviewer #2: No

---

## [Author Response · Author response to Decision Letter 0]

28 Mar 2024

Dear Academic Editor Sascha Köpke

Thank you for considering our manuscript "Feasibility of the “Preventing functional decline in acutely hospitalized older patients (PREV_FUNC)” study – A three-armed randomized controlled pilot trial" for publication in PLOS ONE. We have revised the manuscript based on the corrections suggested by the Academic Editor and the Reviewers. The respond to each comment can be found below. Our revisions in the manuscript can be followed by the “tracked changes” function.

Sincerely,

Linda Sandberg, PhD

Karolinska Institutet 

Department of Neurobiology, Care Sciences and Society 

Division of Physiotherapy, 23100

141 83 Huddinge, Sweden 

Email: linda.m.sandberg@ki.se

Authors’ responses to Editor’s and Reviewers’ comments

Comments from Journal requirements / Academic Editor

Comment

Response

We have carefully reviewed the requirements, corrected the names of the files (page 30), and made minor adjustments to the font used in the title (page 1, line 1-3).

Comment

Response

We appreciate the opportunity to respond to the prompts regarding data availability for our study. Please see our response below. 

Comment

Response

Ethical and Legal Restrictions: We acknowledge the importance of data sharing for scientific transparency and reproducibility. However, we must highlight that our dataset contains coded data from 39 participants. There is a key code that can be used to trace back the data to living individuals. According to the Swedish Archive Law, we need to keep the key code for at least 10 years post publication. The existence of the code means that the data are “pseudonymized personal data” as per European General Data Protection Regulation (GDPR) and cannot be published/shared openly. There is no other way to de-identify the data, given the small sample size, there always remains a risk of potential re-identification of individual participants. The European GDPR law on data protection and privacy imposes restrictions on openly sharing personal data. Our study has also been conducted in compliance with ethical guidelines, and the participants have not given their consent for their data to be made fully public. Sharing the data openly would not only constitute a GDPR breach but also a breach of the Swedish Ethical Approval law that has approved the research on the terms that personal data are protected as per GDPR and as per the conditions that the participants have agreed to. 

For inquiries regarding access to the data, we would like to provide contact information for our Research Data Office at Karolinska Institutet: rdo@ki.se. They will handle requests for data access and ensure compliance with legal and ethical standards.

Comment

Response

Data Availability Statement: In light of the restrictions outlined above, we propose updating the Data Availability Statement as follows:

"The data in the study are pseudonymized (coded) personal data, and European General Data Protection Regulation prohibits us from sharing this completely open. The dataset only includes data from 39 human research participants. Due to the small sample size, there is a risk of identification of individual participants even though the data is de-identified. Data is available upon request, and requests for access to the data can be put to our Research Data Office (rdo@ki.se) at Karolinska Institutet."

We trust that this clarification addresses the concerns raised by the journal regarding data sharing. We remain committed to promoting transparency and scientific integrity while ensuring compliance with legal and ethical standards.

Comment

3. In the online submission form, you indicated that [Data Availability Statement: The data in the study are pseudonymised (coded) personal data, and Swedish legislation prohibits us from sharing this completely open. The dataset only includes data from 39 human research participants. Due to the small sample size risk of identification of individual participants exists if the data is shared, even though data is de-identified. Data is available upon request, and requests for access to the data can be put to our Research Data Office (rdo@ki.se) at Karolinska Institutet.]. 

Response

Please see our response to point number 2a and 2b above. 

Comment

Response

Thank you for addressing this matter. We have now incorporated an Ethics statement in which we have relocated the relevant text from the sections Study design and participants and Procedure (page 6-7, line 130-139 and 144-145). The statement now reads as follows “Ethical approval was obtained from The Swedish Ethical Review Authority Dnr. 2020-06505, and 2021-06788-02. All participants received verbal and written information and informed written consent was obtained before inclusion.” 

Comment

Response

We have corrected the names of the following files

From: S1 File. English translation – Ethical approval 1. 

To: S2 File. Ethical approval 1 – English translation.

From: S2 File. English translation – Ethical approval 2. 

To: S4 File. Ethical approval 2 – English translation.

We will add the following Supporting information files to the list in the manuscript and corrected the names of the files:

From: File Ethical approval 1.pdf 

To: S1 File. Ethical approval 1 – In Swedish.

From: File Ethical approval 2.pdf 

To: S3 File. Ethical approval 2 – In Swedish.

From: 2021-06788-02.pdf (Etikansökan på svenska 2021-06788-02)

To: S5 File. Ethical application – In Swedish.

From: File English transl 2021-06788-02.pdf )

To: S6 File. Ethical application – English translation.

Comment

Response

Corrections have been made in the reference list. Reference 3 has been changed.

Reviewers' comments

Comments

1. Is the manuscript technically sound, and do the data support the conclusions?

Reviewer #1: Yes

Reviewer #2: Yes

2. Has the statistical analysis been performed appropriately and rigorously? 

Reviewer #1: Yes

Reviewer #2: Yes

3. Have the authors made all data underlying the findings in their manuscript fully available?

Reviewer #1: Yes

Reviewer #2: No

4. Is the manuscript presented in an intelligible fashion and written in standard English?

Reviewer #1: Yes

Reviewer #2: Yes

Response: Thank you for taking the time to review the manuscript. Regarding question number 3, data availability, please see our response to Editor’s comments point number 2a and 2b above. 

 

Reviewer #1

Comment

Feasibility of the “Preventing functional decline in acutely hospitalized older patients” study – A three-armed randomized controlled trial

This manuscript described a randomized controlled trial assessing recruitment and retention rates, intervention compliance, acceptability, and scientific feasibility of 3 different exercise programs for acutely ill older inpatients in geriatric units or clinics of 3 different hospitals in Sweden. The authors highlight that few studies have compared different exercise programs in the acute care setting among older patients.

I think overall this is a well-designed assessment and the authors did a very good job of writing up their findings. 

Response

Thank you for reviewing our manuscript, and for your positive comments regarding the manuscript. 

Comment

One question I have relates to inpatients with mild cognitive impairment (MCI). Other studies have shown that at least 1/3 of older inpatients have some level of cognitive impairment, and that patients with MCI are very limited in terms of physical function and mobility. Have the authors considered the impact of their interventions for patients with MCI? It seems these interventions are helpful for many but not all patients. Also, what happens when a patient develops delirium during hospitalization? How will the authors address this issue in the later RCT? The authors should explain why they are not assessing cognition at baseline using a standardized validated tool such as the Mini-cog. This is important not only for confirming the patient can provide written consent, but also for confirming cognition status. I think this piece of data should be prioritized during recruitment and enrollment for the later RCT and can be used to better understand the results of the study. If it is possible to include some patients with MCI, that would ultimately provide more evidence to present related to this trial. Alternatively, if the exercise programs are not appropriate for patients with MCI that should be clearly articulated and explained.

Response

Thank you for allowing us to address this important issue. Our inclusion criteria did not exclude participants with cognitive impairment; rather, they emphasized the participants’ ability to communicate and collaborate with the research staff. As stated in the Procedure section (page 7, line 145-148), “Consent was obtained by research staff with sensitivity, meaning that no one was enrolled in the study if they appeared to have difficulty understanding what was meant by informed consent or expressed discomfort regarding participating.” 

However, we concur with the Reviewer that the absence of an objective measure of cognitive function represents a limitation in our study. To address this concern, we have included the following sentences in the discussion (page 22, line 427-432). “Our study did not incorporate a measure of cognitive function. Consequently, we cannot definitively conclude whether the intervention is feasible for older adults with cognitive impairment.”

Comment

The first reference was cited twice – reference 1 and reference 3 in the bibliography.

Response

Thank you for noticing this. This has been corrected (page 4, line 62 and 64, and page 25).

Reviewer #2

Comment

This paper reports preliminary results from a feasibility study of in-hospital exercise. The manuscript is clearly written with intervention and analytical details well-documented. The major limitation is the small sample size. However, the authors clarified that the purpose of the study is just to examine the feasibility rather than intervention effectiveness. I only have a few minor suggestions:

Response

Thank you for your careful revision of our manuscript, and for helping us to further improve it. Please, see our responses to your comments below.

Comment

1. The study excludes individuals living in nursing homes. It would be helpful if the authors could provide an explanation for this exclusion, given that the intervention is conducted in hospitals.

Response

Thank you for allowing us to clarify this issue. To address this concern, we have included the following sentences in the discussion (page 22, line 433-438). “Furthermore, we excluded individuals living in nursing homes for two specific reasons. Firstly, a comparable exercise program to the simple exercise program had already been tested in nursing homes as part of a previous study [32]. Secondly, considering that participants in the Simple exercise program will be encouraged to continue their exercises at home during the full-scale trial, it was deemed more appropriate to include only those living in their own homes.”

---

## [Decision Letter · Decision Letter 1]

15 May 2024

Feasibility of the “Preventing functional decline in acutely hospitalized older patients (PREV_FUNC)” study – A three-armed randomized controlled pilot trial

PONE-D-23-42196R1

Dear Dr. Sandberg,

We’re pleased to inform you that your manuscript has been judged scientifically suitable for publication and will be formally accepted for publication once it meets all outstanding technical requirements.

Kind regards,

Sascha Köpke

Academic Editor

PLOS ONE

Reviewer's Responses to Questions

**Comments to the Author**

1. If the authors have adequately addressed your comments raised in a previous round of review and you feel that this manuscript is now acceptable for publication, you may indicate that here to bypass the “Comments to the Author” section, enter your conflict of interest statement in the “Confidential to Editor” section, and submit your "Accept" recommendation.

Reviewer #1: All comments have been addressed

Reviewer #2: All comments have been addressed

2. Is the manuscript technically sound, and do the data support the conclusions?

Reviewer #1: Yes

Reviewer #2: Yes

3. Has the statistical analysis been performed appropriately and rigorously? 

Reviewer #1: Yes

Reviewer #2: Yes

4. Have the authors made all data underlying the findings in their manuscript fully available?

Reviewer #1: Yes

Reviewer #2: No

5. Is the manuscript presented in an intelligible fashion and written in standard English?

Reviewer #1: Yes

Reviewer #2: Yes

6. Review Comments to the Author

Reviewer #1: The comments were adequately addressed. The authors also provided adequate documentation regarding data privacy laws in Sweden.

Reviewer #2: The authors have satisfactorily addressed my previous comments. The current version is good for publication.

7. PLOS authors have the option to publish the peer review history of their article (what does this mean?). If published, this will include your full peer review and any attached files.

Reviewer #1: **Yes: **Christine M. Loyd

Reviewer #2: No

---

## [Editor Report · Acceptance letter]

27 May 2024

PONE-D-23-42196R1 

PLOS ONE

Dear Dr. Sandberg, 

I'm pleased to inform you that your manuscript has been deemed suitable for publication in PLOS ONE. Congratulations! Your manuscript is now being handed over to our production team.

Kind regards, 

on behalf of

Professor Sascha Köpke 

Academic Editor

PLOS ONE